# Effect of Dietary Restriction on Gut Microbiota and Brain–Gut Short Neuropeptide F in Mud Crab, *Scylla paramamosain*

**DOI:** 10.3390/ani14162415

**Published:** 2024-08-20

**Authors:** Chenchang Bao, Yanan Yang, Haihui Ye

**Affiliations:** 1School of Marine Sciences, Ningbo University, Ningbo 315832, China; yangyanan@nbu.edu.cn; 2School of Fisheries, Jimei University, Xiamen 361021, China; hhye@jmu.edu.cn

**Keywords:** gut microbiota, brain–gut sNPF, feed deprivation, starvation, mud crab

## Abstract

**Simple Summary:**

While the gut microbiota and gut–brain sNPF are known to play crucial roles in diet restriction, research on their response to feed deprivation and starvation stress remains limited. This work investigated the alterations in the gut microbiota of juvenile mud crabs under feed deprivation and starvation conditions and identified the location of sNPF in the brain and gut. This suggests a potential interplay between diet restriction, gut microbiota, and gut–brain sNPF. The increased expression levels of sNPF across various levels of feed deprivation further support this notion. This study suggests that sNPF may regulate digestive activities and immunity through interactions with the gut microbiota. These findings provide novel insights into the dynamic interplay between the gut microbiota and sNPF in response to diet restriction, emphasizing their crucial roles in physiological adaptation to starvation stress.

**Abstract:**

Aquatic animals frequently undergo feed deprivation and starvation stress. It is well-known that the gut microbiota and the gut–brain short neuropeptide F (sNPF) play essential roles in diet restriction. Therefore, investigating the responses of the gut microbiota and sNPF can enhance our understanding of physiological adaptations to feed deprivation and starvation stress. In this study, we examined the alterations in the gut microbiota of juvenile mud crabs under feed deprivation and starvation conditions. The results reveal differences in the richness and diversity of gut microbiota among the satisfied, half food, and starvation groups. Moreover, the microbial composition was affected by starvation stress, and more than 30 bacterial taxa exhibited significantly different abundances among the three feeding conditions. These results indicate that the diversity and composition of the gut microbiota are influenced by diet restriction, potentially involving interactions with the gut–brain sNPF. Subsequently, we detected the location of sNPF in the brains and guts of mud crabs through immunofluorescence and investigated the expression profile of sNPF under different feeding conditions. The results suggest that sNPF is located in both the brains and guts of mud crabs and shows increased expression levels among different degrees of diet restriction during a 96 h period. This study suggested a potential role for sNPF in regulating digestive activities and immunity through interactions with the gut microbiota. In conclusion, these findings significantly contribute to our understanding of the dynamic changes in gut microbiota and sNPF, highlighting their interplay in response to diet restriction.

## 1. Introduction

Aquatic animals often undergo feed deprivation and starvation due to changes in environmental factors [1]. Sometimes, short−term starvation may be part of a strategy to mitigate the deterioration of water quality in aquaculture [2]. Starvation not only affects the nutritional and health status of the animals but also the microbial composition in the intestine [3]. Starvation changes the composition of the microbial community, disturbs the intestinal immune system, and stimulates inflammatory responses in aquatic animals, such as fish and shrimp [4,5,6,7]. It is well−known that the intestinal microbiota plays an important role in intestinal development and physiology as well as in general development, growth, and health [8].

There is emerging evidence of important links between the gut microbiome and the brain [9]. Neuropeptides are highly enriched in the brain and involved in most physiological processes, including feeding, digestion, immunity, energy homeostasis, and reproduction [10,11,12,13,14]. Additionally, neuropeptides have an impact on the composition and function of the gut microbiota and its relevance to the brain–gut axis [15,16]. The neuropeptide Y (NPY) family, as the classic brain–gut peptide family, plays a pivotal role in the regulation of food intake among invertebrates and vertebrates [17]. In addition, allatostatin–A (AST–A), AST–B, AST–C, CCHamide, diuretic hormone 31 (DH31), and tachykinin also all function as brain–gut peptides in arthropods [18,19,20]. Short neuropeptide F (sNPF), a member of the NPY family, is 6–14 amino acid residues in length, features a conserved C-terminus, RxRFamide, and has been widely identified from arthropods. In insects, sNPF plays an essential role in the feeding state and the starvation process [11,21,22]. As a brain–gut peptide, sNPF inhibits digestive activity in the American cockroach [11] and regulates olfactory sensitivity to promote feeding behavior in flies during starvation [17,18]. Additionally, sNPFs in the midgut might function as enterogastric peptides and regulate the feeding behavior of the fly [23]. To date, the effects of feed deprivation and starvation on the sNPF of insects have been studied well, while little research has been conducted on aquatic animals, especially on the crustacean brain–gut microbiome.

The mud crab *Scylla paramamosain*, a commercially important crustacean species, has been widely cultivated in the Asia–Pacific region [24]. Crabs frequently suffer from starvation, restricted food availability, or deprivation due to environmental changes in their habitat and to climate conditions in both natural or aquaculture settings [25,26,27]. Thus, exploring their gut microbiota responses would improve our understanding of the crab’s physiological adaptations to feed deprivation and starvation. Given the crucial regulatory role of the sNPF in feed deprivation and starvation, along with previous observations in the brain–gut microbiome, we hypothesize that both feed deprivation and starvation can influence the structure of microbial communities and the expression of brain–gut sNPF.

In this study, we aimed to evaluate the changes in the composition of the microbial community and to investigate the sNPF response in the mud crab under feed deprivation and starvation conditions. We also studied the correlation between the sNPF and the gut microbiome. Our findings provide new insights into the interplay between the gut microbiota and brain–gut sNPF in response to feed deprivation and starvation stress in the mud crab, *S. paramamosain*.

## 2. Materials and Methods

### 2.1. Sample Collection

Juvenile crabs (C2 stage, weight 17.26 ± 1.32 g) were used for feeding tests. The juvenile crabs were independently reared in round plastic culture vessels (diameter: 5 cm; height: 10 cm) to avoid cannibalism, with saline seawater at a concentration of 22 ± 1 ppm and a temperature of 26 ± 87 0.5 °C, and under an alternated light/dark cycle of 12L:12D. A daily 100% water exchange was carried out, and crabs were fed once daily at 8:00 am by hand. Juvenile crabs (n = 20) were fed on the weighed and chopped Manila clam *Ruditapes philippinarum*; to determine the maximum daily food consumption, the remaining food was removed, dried, and weighed after 24 h.

Three different feeding conditions—namely, enough–ood (EF), 1/2-food (HF), and starvation (S) groups—were designed to analyze the microbial community in the gut of the juvenile crabs. A total of 252 juvenile crabs were divided into three groups. After 72 h, 17 to 20 juveniles from each feeding condition were euthanized in an ice slurry for approximately 1 min, surface sterilized by dipping in 70% ethanol for 15 s, and then placed on sterile Petri dishes for dissections. Guts were dissected from the juvenile crabs and placed into 2 mL sterile tubes for dissection. All samples were stored at −80 °C until DNA extraction. Each feeding condition was performed with three biological replicates.

To investigate the function of sNPF in different feeding conditions, 96 juvenile crabs were divided into four groups: enough-food, 1/2-food, 1/4-food, and starvation. At 6, 12, 24, 48, 72, and 96 h, three individuals from each condition were randomly sampled to analyze the expression of sNPF. All samples were stored at −80 °C until RNA extraction.

For the immunofluorescence assay, mud crabs (weight 314–379 g) were purchased from the local fish market in Xiamen, China. Prior to dissections, the crabs were anesthetized on ice for 15 min, and brain and gut were collected from the crabs.

### 2.2. High-Throughput Sequencing and Bioinformatic Analysis

DNA from the gut bacteria was extracted using the QIAamp-DNA stool minikit (Qiagen, Hilden, Germany) according to the manufacturer’s instructions. After assessing the DNA quality using a Nanodrop ND2000 UV–Vis spectrophotometer (Thermo Scientific, Wilmington, NC, USA) and agarose gel electrophoresis, the V3–V4 region of the 16S rRNA gene was amplified with the barcode-fusion forward primer 338F (5′-ACTCCTACGGGAGGCAGCA-3′) and the reverse primer 806R (5′-GGACTACHVGGGTWTCTAAT-3′) [28].

The PCR protocol was performed in 50 µL mixtures containing 10 µL of 5 × *TransStart^®^ FastPfu* (TransGen, Beijing, China), 4 µL of 2.5 mM dNTPs, 1 µL of each primer (5 µM), 15 ng of template DNA, and 1 µL of *TransStart^®^ FastPfu* (TransGen, Beijing, China) DNA Polymerase. The thermal profile for PCR was 30 s at 98 °C, 25 cycles of 30 s at 98 °C, 15 s at 48 °C, and 30 s at 72 °C, followed by a post-amplification extension of 7 min at 72 °C. Amplicons were extracted from 2% agarose gel and purified using the AxyPrep DNA Gel Extraction Kit (Axygen Biosciences, Union City, CA, USA). Subsequently, the amplicons were sequenced on an Illumina PE300 platform (San Diego, CA, USA) from a commercial company (Shanghai Majorbio Bio-pharm Biotechnology Co., Ltd., Shanghai, China). The raw reads of gut bacteria were uploaded to the NCBI Sequence Read Archive (accession number: PRJNA1069600).

The raw sequencing of data was in FASTQ format. Paired-end reads underwent preprocessing with Trimmomatic software [29] to detect and remove ambiguous bases (N). After trimming, the paired-end reads were assembled using FLASH software [30]. Quality filtering, chimeric sequence removal, and merging of paired-end reads were performed using the DADA2 plugin [31]. The sequences were then clustered into operational taxonomic units (OTUs) at the 97% sequence similarity level using UPARSE software v7.1 [32]. Bioinformatic analysis of the gut microbiota was carried out using the Majorbio Cloud platform (https://cloud.majorbio.com, accessed on 4 January 2024). For alpha-diversity analysis, we purified the OTUs to several indicators, including OTU rank curves and counted indices such as Chao, ACE, Shannon, and Simpson. Principal component analysis (PCA) and linear discriminant analysis (LDA) and effect size (LEfSe) were applied to quantify and compare the compositional differences between microbial communities.

### 2.3. RNA Extraction and cDNA Synthesis

Total RNA was extracted from the juvenile crabs using Trizol RNA isolation reagent (Invitrogen, Carlsbad, CA, USA) according to the manufacturer’s instructions. The quality and concentration of RNA were monitored by using agarose gel electrophoresis and NanoOne spectrophotometer. First-strand cDNA was synthesized from 2 μg of total RNA using the PrimeScript^TM^ RT reagent kit with gDNA Eraser (TaKaRa Bio Inc., Otsu, Shiga, Japan).

### 2.4. Whole-Mount Immunohistochemistry

Given that antisera produced by mature sNPF peptides with C-terminal motif “RLRFamide” are likely to cross-react with the RFamide neuropeptides [33], the antiserum produced by peptide “PQRLRWamide” (a kind of gift from Pro. Veenstra at the University of Bordeaux) was selected for whole-mount immunohistochemistry. The “RLRWamide” antiserum had been verified to label the same neurons as the sNPF precursor antiserum [33,34]. The specificity of the antiserum was tested by immunocytochemistry after preincubation with the sNPF peptide used for immunization [33].

Whole-mount immunofluorescent staining was performed based on the previously described method [35,36]. The brain and midgut were fixed in 4% paraformaldehyde (PFA) at 4 °C overnight. After fixation, the samples were rinsed five times at 1-h intervals in a PBS solution containing 0.3% Triton X-100 (PBST) and incubated in a 1:4000 dilution of the sNPF antibody (“RLRWamide” antiserum) at 4 °C for 48 h. Following incubation in the sNPF antibody, tissues were rinsed five times at 1-h intervals in PBST and then incubated in a 1:400 dilution of goat anti-rabbit immunoglobulin G (IgG) labeled with Alexa Fluor 488 (Beyotime, Shanghai, China) at 4 °C for 24 h. Both antibodies were diluted by PBST containing 10% Bovine Serum Albumin (BSA, Yuanye Bio-Technology Co., Ltd., Shanghai, China). Before visualization, tissues were incubated in 4′, 6-diamidino-2-phenylindole (DAPI, Invitrogen, Carlsbad, CA, USA) for 30 min to stain the nucleus, and then incubated in scale regent for 72 h to render the tissues transparent [37]. Finally, stained tissues were mounted in Antifade Polyvinylpyrrolidone Mounting Medium (Beyotime, Shanghai, China) and imaged using laser confocal fluorescence microscopy (LSM 880, Zeiss).

### 2.5. Expression Profile of sNPF in Different Feeding Conditions

Fluorescence quantitative real−time PCR (qRT−PCR) was conducted to detect the temporal expression profile of sNPF under different feeding conditions. Gene-specific primers for sNPF and the housekeeping gene *β-actin* were described in previous study [14]. The qRT-PCR was carried out in 20 μL mixtures, containing 10 μL of SYBR Premix Ex Taq^TM^ (TaKaRa), 0.4 μL of each primer (10 mM), 2 μL of the cDNA, and 0.4 μL of ROX^TM^ Reference Dye (TaKaRa). The profile for qRT-PCR was 30 s at 95 °C for 1 cycle; followed by 5 s at 95 °C, 30 s at 55 °C, and 30 s at 72 °C for 40 cycles. QRT-PCR measurements were performed using the Applied Biosystems 7500 Real–time PCR System (Carlsbad, CA, USA) version 2.4 software. All experiments were performed with three biological replicates and each biological replicate performed three technical replicates.

### 2.6. Statistical Analyses

In microbiome analysis, the nonparametric factorial Kruskal–Wallis (KW) rank-sum test and the (unpaired) Wilcoxon rank-sum test were utilized to recognize the most differently abundant taxa among the three groups (LDA score > 2, *p*-value < 0.05). sNPF expression relative to the control was determined by the 2^–ΔΔCt^ method. All data are presented as the mean ± SD. Statistical analyses were performed using SPSS 20 software. Data were normally distributed, as assessed with normality tests (Shapiro–Wilk test). After testing the homogeneity of variances with Levene’s test, data were subjected to one−way analysis of variance (ANOVA) with Scheffé’s method for post hoc analysis. *p*-values less than 0.05 were considered statistically significant.

## 3. Result

### 3.1. Summary of Sequencing

A total of 370,583 sequences were obtained from the microbial samples of the crab gut after quality filtering and chimera removal. The mean number of sequences per samples was 414 base pairs (bps) (Appendix A). The rarefaction curves for all samples tended to plateau, indicating the sequencing of all samples reached saturation and was sufficient to produce stable estimates of alpha diversity or species richness (Appendix A). A total of 551 operational taxonomic units (OTUs) were identified, with 373 OTUs (73.72% of S, 79.36% of HF, and 86.54% of EF) shared among the three gut samples (Appendix A).

### 3.2. Richness and Diversity Analysis

The species richness (mean number of OTUs) for ACE showed 21.59 ± 1.42 in the S group, 21.03 ± 1.48 in the HF group, and 16.64 ± 2.29 in the EF group, respectively (Table 1). The ACE index was significantly higher in the S and HF group compared to the EF group (*p* < 0.05). However, no significant differences were observed in the Chao index among the three groups. Additionally, there were no significant differences in diversity across the groups. The investigation of beta diversity indicated the formation of three clusters of gut libraries, with the HS, S, and EF groups forming individual clusters (Figure 1). The observed separation among the different food intake groups suggests that feeding significantly influences gut bacterial communities.

### 3.3. Microbiome Composition

A taxonomic summary of the mud gut microbiome is presented in Figure 2. We identified 10 phyla, with the four most dominant communities being Proteobacteria, Bacteroidota, Firmicutes, and Actinobacteriota (Figure 2a). The relative abundance of dominant phyla and genera varied among the different feeding conditions (Figure 2b,c). The HF and EF groups exhibited a significantly higher abundance of Proteobacteria compared to the S group (*p* < 0.05), while no significant difference in Bacteroidota, Firmicutes, and Actinobacteriota was observed (Figure 2b). The top 20 high genera with the highest abundance were selected for multiple comparisons (Figure 2c). The most dominant genera, including *Ruegeria*, unclassified_f_Rhodobacteraceae, *Raoultella*, *Tenacibaculum,* and *Vibrio*, remained consistent across different feeding modes. However, the unclassified_f_Halieaceae and norank_f_Flavobacteriaceae showed a significantly higher abundance in the S group than in the EF and HF groups (*p* < 0.05).

An LEfSe analysis (LDA score > 2) was conducted to identify key differential taxa in the gut microflora of juvenile crabs (Figure 3). As shown in cladograms and the LDA score, there were 29 biomarkers in the S group, Dependentiae and Campilobacterota were phylum-level biomarkers, and *Marinococcus*, *Bdellovibrio*, *Waddlia*, *Aureisphaera*, *Iamia*, *Limnobacter*, *Marinifilum*, *Cycloclasticus,* and *Micrococcus* were genus–level biomarkers in the S group. According to the LEfSe analysis, Pseudomonadales was the only biomarker identified in the HF group. In the EF group, we found that Campilobacterota (from phylum to genus) and *Gilvibacter* were biomarkers.

### 3.4. Localization of sNPF in the Brain and Midgut

sNPF antiserum labeling was observed throughout the brain, with robust immunoreactivity detected in olfactory lobes (OLs) and the adjacent cell cluster 9 (Figure 4a). Approximately one hundred labelled neurosecretory cells were observed in cluster 9. Additionally, sNPF antiserum labeling was evident throughout the gut, exhibiting strong immunoreactivity (Figure 4b). The localization of sNPF indicates that it functions as a brain–gut neuropeptide in *S. paramamosain*.

### 3.5. Effects of Different Feeding Conditions on sNPF Expression

The effects of different feeding conditions on sNPF precursor transcript expression were tested using qRT-PCR across four groups. After 24 h, a significant increase in sNPF expression was observed under feed restriction conditions. In the enough-food group, the expression levels of sNPF remained essentially unchanged throughout the experiment. Conversely, in the 1/2-food group, the expression levels of sNPF were significantly up−regulated by 6–fold, 2–fold, 5-fold and 4-fold from 24 to 96 h. Similarly, in the 1/4-food group, the sNPF expression levels showed significant up−regulation by 7-fold, 6-fold, 6-fold, and 6-fold from 24 to 96 h. Starvation resulted in a significant increase in sNPF expression by 5-fold and 3-fold at 24 and 48 h, respectively. However, the expression levels of sNPF showed no significant differences between the enough-food and starvation groups at 72 and 96 h (Figure 5). These results suggest that sNPF is related to dietary restriction in mud crab.

## 4. Discussion

The assembly of the gut microbial community is influenced by various factors, including physiology, the gut–brain axis, genetic factors, and the living environment [15,38,39]. In turn, the gut microbiota also exerts significant effects on host development, nutrient digestion, and absorption, as well as the immune system [40,41]. The gut microbial ecosystem of crustaceans undergoes changes in response to diet, which are sometimes linked to variations in host performance [42]. In this study, we identified that both feed deprivation and starvation directly affect the diversity of gut microbiota in juvenile mud crabs. This leads to the conclusion that the gut microbiota in aquatic animals is correlated with dietary restrictions. Previous research has also observed changes in gut microbiota diversity due to starvation in species such as the loach *Paramisgurnus dabryanus* [43] and juvenile blunt snout bream *Megalobrama amblycephala* [44], as well as black tiger shrimp *Penaeus monodon* [42].

The composition of microflora in the gut of juvenile mud crabs was dominated by Proteobacteria, Bacteroidota, Firmicutes, and Actinobacteriota. Similar results were reported in studies conducted on the gut of the horseshoe crab *Tachypleus tridentatus* [45] and the Pacific white shrimp *Litopenaeus vannamei* [46]. However, in the gut of adult mud crabs, Tenericutes, Proteobacteria, Bacteroidetes, Firmicutes, and Fusobacteria are the dominant phyla [16]. This dissimilarity may be attributed to the different diets of mud crabs, which are influenced by farming conditions and development stages [47]. The abundance of Proteobacteria was related to dietary interventions [48]. Our data showed that the relative abundance of Proteobacteria was lower in the starvation group compared to the feeding groups, which aligns with findings in the Asian seabass *Lates calcarifer* [49]. It has been reported that Proteobacteria were associated with unstable gut microbiota, energy instability, and inflammation [4,50]. Furthermore, starvation primarily increased the relative abundance of unclassified Halieaceae and norank Flavobacteriaceae, which possess diverse metabolic capabilities [51,52]. Current research suggests that starvation significantly alters the intestinal microbial structure of mud crabs, potentially impacting their energy homeostasis, inflammation, and metabolic capabilities. This finding is consistent with previous observations [53] that starvation induces changes in energy, metabolism, and the stability of the intestinal microbiota, ultimately leading to a substantial decrease in intestinal microbiota.

It has been suggested that some microbial members are more sensitive to disturbance, even though the overall community remains resistant [54]. In line with this notion, we observed that more than 30 bacterial taxa exhibited significantly different abundances among the three feeding conditions. Campilobacterota showed a higher abundance in the enough-food group, playing important roles in maintaining the normal functions of the host intestinal environment [55]. This taxon is commonly found in the gut microbiota of aquatic invertebrates, including adult mud crabs (*S. paramamosain*). Additionally, the genus *Gilvibacter* has been reported as a biomarker associated with early stages (zoea 2) in *P. vannamei* larvae [56] and was also a biomarker in the enough-food group of juvenile crabs in the present study. The absence of Campilobacterota and *Gilvibacter* in the 1/2-food and starved groups may disrupt the gut bacterial community in feed-deprived juvenile crabs. Similar observations of gut bacterial community disequilibrium have been reported in shrimp [6]. The higher abundance of Pseudomonadales in the 1/2-food group indicates an indirect but pronounced role in digestion. A high abundance of Pseudomonadales has also been reported in juvenile rainbow trout *Oncorhynchus mykiss* fed plant-derived dietary proteins [57]. Notably, the genus *Waddlia*, a Chlamydia-related bacterium, is considered a pathogen infecting a wide variety of hosts, including mammalian, fish, and insect cells [58]. Similarly, the class Babeliae of the phylum Dependitae was more abundant in the starved crabs, most of which are isolated from pathogens [59]. Therefore, the colonization of Waddlia and Babeliae may pose a threat to the health of starved crabs. Our results indicated that feed deprivation and starvation can disrupt the gut bacterial community in juvenile crabs (*S. paramamosain*). It has been reported that neuropeptides exhibit immune regulatory functions and impact the composition and function of the gut microbiota [15,60]. The neuropeptide sNPF has been found to increase the expression of antimicrobial peptides in response to the bacteria analog LPS or virus analog Poly (I:C) [13]. Thus, we speculate that the microbial composition in the gut of mud crabs may be regulated by sNPF during feed deprivation and starvation.

The brain–gut–microbiome axis demonstrates bidirectional communication [15]. Gut signals, including those from gut microbiota and gut hormones, can be transmitted from the gut to the brain, thereby regulating brain function and related behaviors. Conversely, neuroendocrine signals can travel from the brain to the gut [15,61]. Neuropeptides play a crucial role in this axis. As a classic gut–brain peptide, NPY has been shown to regulate gut microbiota [15,62]. In *Drosophila*, the enterocytes of the midgut respond to dietary and microbiota-derived protein levels, regulating the expression of the neuropeptide CNMamide, which communicates with the brain [63].

The neuropeptide sNPF is a well-known brain–gut peptide in insects [11,33,64]. In *Drosophila*, sNPF immunoreactive staining was detected in the odorant receptor neurons of the antennae [33,41,65]. In the present study, whole-mount immunohistochemistry revealed that sNPF is present in both the brain and gut of mud crabs, suggesting that sNPF serves as a brain–gut peptide in mud crabs. The OLs, critical components of the central olfactory system, have been extensively studied in various crustaceans, including lobsters, crayfish, and crabs (review in Harzsch and Krieger [66]). The OLs exhibit a glomerular structure on each side of the deutocerebrum and receive the primary afferent endings of the chemoreceptors located on the antennae [67]. Consequently, the OLs are primarily responsible for analyzing the odorant quality [66]. Cell cluster 9 contains a large number of small neurosecretory cells, and the primary neurites directed toward the OLs [67]. Here, the strong sNPF immunoreactivity observed in OLs and cell cluster 9 suggests that sNPF acts as a neuromodulator in mud crabs. As a brain–gut peptide, sNPF has also been identified in enteroendocrine cells in the gut, where it is involved in regulating digestive activity [11,68] and feeding behavior [19,68]. In this research, we detected hundreds of sNPF immunoreactive cells in the gut, indicating that sNPF may interact with gut microbiota to regulate digestive activities and immunity in mud crabs, similar to another RF-amide neuropeptide, NPY [15].

There is evidence that sNPF plays an essential role in the feeding states and starvation processes in insects [21,22,69]. Our findings indicate that the sNPF transcript increased by different degrees during a 96-h period of dietary restriction. Similar results have been reported in insects; for instance, the levels of sNPF transcript increased due to starvation in *B. dorsalis* [22], and sNPF expression in neurons increased, leading to enhanced food-searching behavior in *D. melanogaster* [21]. Thus, we hypothesized that sNPF is related to dietary restriction in mud crabs. It has been reported that gut microbiota regulates neuropeptides, likely by influencing enteroendocrine cells in epithelial cells [15,60]. We speculated that an imbalance in gut microflora may disrupt enteroendocrine cells, affecting the expression of sNPF after 72 h of feed deprivation. Therefore, it is reasonable to conclude that, as a brain–gut peptide, sNPF may be involved in controlling digestive activities and immunity through interaction with gut microbiota.

## 5. Conclusions

In conclusion, this study reveals significant alterations in the diversity, structure, and function of the microbiota, as well as changes in the expression of brain–gut sNPF in the gut of juvenile mud crabs subjected to feed deprivation and starvation. The observed changes in gut microbiota are associated with the expression of sNPF during these conditions, with sNPF localized in both the brain and gut of mud crabs. Furthermore, our findings suggest a relationship between sNPF and dietary restriction in mud crabs, indicating its potential role in digestive activities and immunity through interactions with gut microbiota. Nevertheless, further research is needed to elucidate the complex interactions between gut microbiota and sNPF during feed deprivation and starvation, particularly in the context of the presence or absence of sNPF. These findings significantly enhance our understanding of the dynamic changes in gut microbiota and sNPF, highlighting their interplay in response to dietary restriction.

## Figures and Tables

**Figure 1 animals-14-02415-f001:**
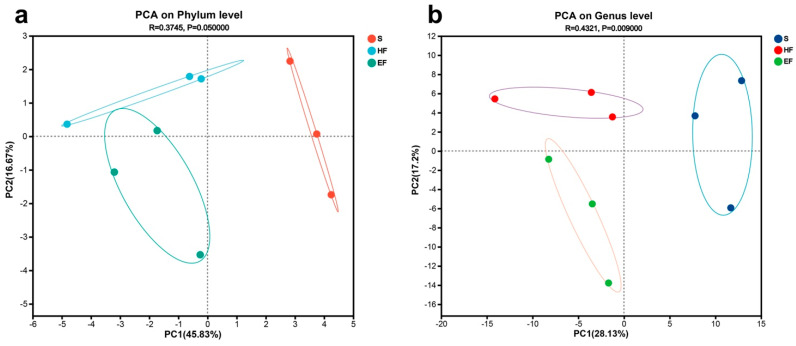
Principal component analysis of the gut microbiome (both at the phylum (**a**) and genus (**b**) level) in satiated, 1/2-food, and starvation groups. The X and Y axes represent the first and second principal coordinates, respectively. The percentage value in the axis label represents the contribution of the corresponding coordinate to the sample variance, indicating how much this principal component extracts from the original information. S, starvation group; HF, 1/2-food group; EF, enough-food group.

**Figure 2 animals-14-02415-f002:**
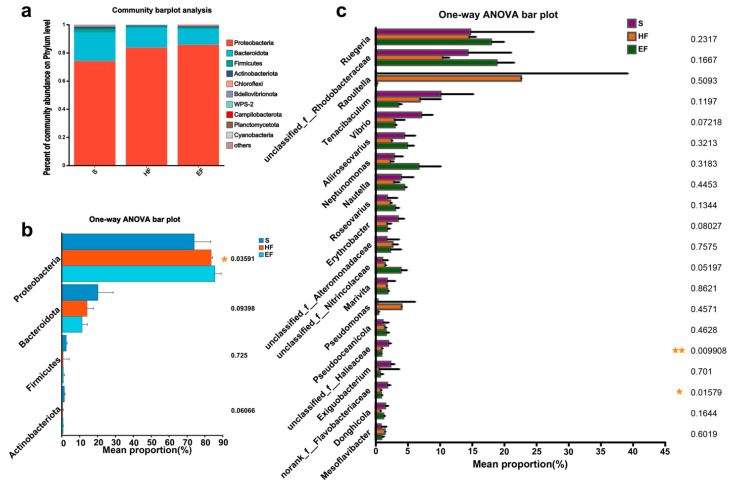
Relative abundances of dominant phyla and comparison of bacterial abundances in the gut at phylum and genus levels. (**a**) Relative abundances of dominant phyla from all samples based on 16S rRNA gene amplicon sequencing data. Unclassified phyla with relative abundances lower than 1% were assigned as “others”. (**b**) Comparison of the four dominant bacterial abundances in the gut at phylum levels. (**c**) Comparison of the top 20 bacterial abundances in the gut at genus levels. The vertical axis represents the name of microbial community, and the column length corresponding to the species represents the average relative abundance of the species in each group. * 0.01 < *p* ≤ 0.05, ** 0.001 < *p* ≤ 0.01. S, starvation group; HF, 1/2-food group; EF, enough-food group.

**Figure 3 animals-14-02415-f003:**
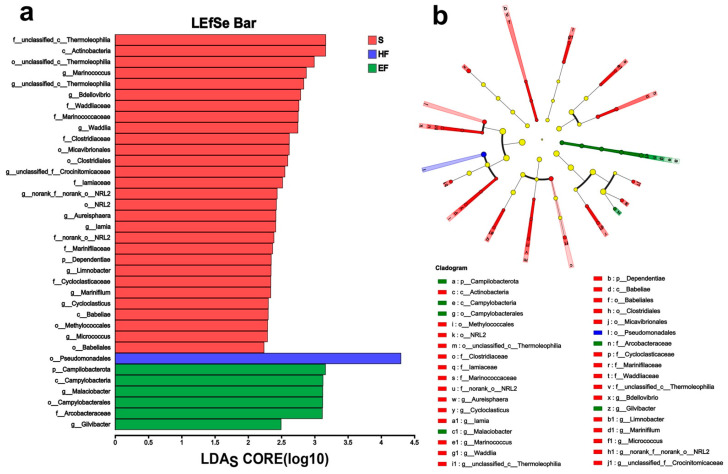
LEfSe analysis of gut microbiome from the three feeding modes. (**a**) Bar chart showing the LDA scores of bacterial taxa identified by LEfSe analysis (taxa with *p* < 0.05 and LDA >2 are shown). (**b**) Cladogram showing the phylogenetic relationships of bacterial taxa revealed by LEfSe. S, starvation group; HF, 1/2-food group; EF, enough-food group.

**Figure 4 animals-14-02415-f004:**
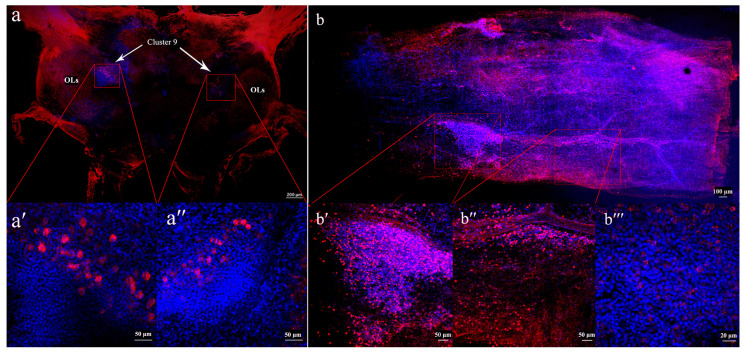
Localization of sNPF immunoreactivity in the brain and midgut of *S. paramamosain*. Red indicates the brain (**a**,**b**) and the midgut of *S. paramamosain*, respectively. Blue indicates cell nuclei labelled by DAPI. OLs, olfactory lobes. (**a′**,**a″**,**b′**,**b″**) Photomicrographs 20× magnification; (**b‴**) Photomicrographs 40× magnification.

**Figure 5 animals-14-02415-f005:**
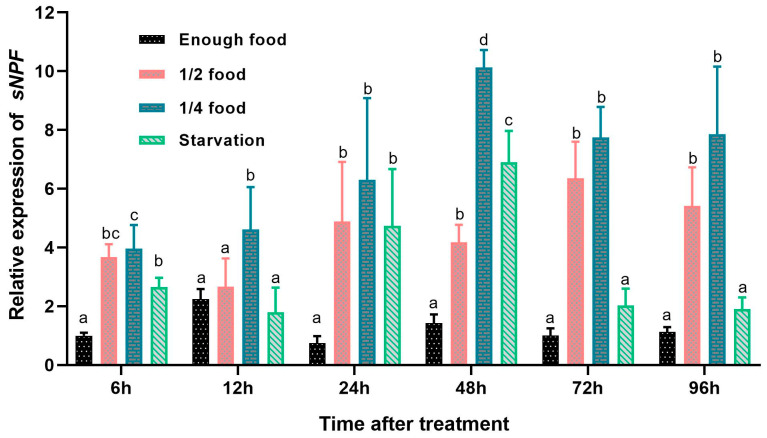
Effects of different feeding conditions on sNPF expression. The relative expression levels of sNPF are shown as the means ± SD (n = 3). Letters above the bars mean with the same letter did not significantly differ, and different letters indicate significant differences (*p* < 0.05) at independent treat time.

**Table 1 animals-14-02415-t001:** Richness and alpha diversity indices for gut microbes in juvenile *S. paramamosain*.

Sample	Richness Estimators		Diversity Indices	
	Chao	ACE	Shannon	Simpson
S	21.33 ± 1.16 ^a^	21.59 ± 1.42 ^a^	0.78 ± 0.07 ^a^	0.59 ± 0.01 ^a^
HF	18.67 ± 23.51 ^a^	21.03 ± 1.48 ^a^	0.52 ± 0.22 ^a^	0.73 ± 0.13 ^a^
EF	16.67 ± 2.31 ^a^	16.64 ± 2.29 ^b^	0.53 ± 0.09 ^a^	0.75 ± 0.05 ^a^

Note: Different lowercase letters (^a,b^) indicate significant differences among different feeding modes (*p* < 0.05). S, starvation group; HF, 1/2-food group; EF, enough-food group.

## Data Availability

The datasets used and analyzed during the current study are available from the corresponding author.

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
