# Peer review of "Effect of Dietary Restriction on Gut Microbiota and Brain–Gut Short Neuropeptide F in Mud Crab, Scylla paramamosain"

_animals, 2024, doi:10.3390/ani14162415_

Round 1

Reviewer 1 Report

Comments and Suggestions for Authors

 Several issues for this paper  should be addressed:

 1.     In the Introduction section, more references of brain-gut peptide should be cited. Apart from NPY family, as I know, many neuropeptides, such as AST-A, AST-B, AST-C, DH31, tachykinin and CCHamide, located in arthropod gut. The authors should cite more articles: Ren, Guilin R., et al. PloS one 10.7 (2015): e0133017.; Wu, Kai, et al. Frontiers in physiology 11 (2020): 191.; Veenstra, Jan A., et al. Cell and tissue research 334 (2008): 499-516.; Wegener, Christian, and Jan A. Veenstra. "Chemical identity, function and regulation of enteroendocrine peptides in insects." Current opinion in insect science 11 (2015): 8-13.

2.     The Discussion is not sufficient. Especially the discussion of “brain-gut-microbiome” axis. References: Holzer, Peter, and Aitak Farzi. "Neuropeptides and the microbiota-gut-brain axis." Microbial endocrinology: the microbiota-gut-brain axis in health and disease (2014): 195-219.; Martin, Clair R., et al. "The brain-gut-microbiome axis." Cellular and molecular gastroenterology and hepatology 6.2 (2018): 133-148.; Kim, Boram, et al. "Response of the microbiome–gut–brain axis in Drosophila to amino acid deficit." Nature 593.7860 (2021): 570-574.; Lerner, Aaron, Sandra Neidhöfer, and Torsten Matthias. "The gut microbiome feelings of the brain: a perspective for non-microbiologists." Microorganisms 5.4 (2017): 66.

3.     Authors should write consistently “mud crabs”, “the mud crabs” throughout the manuscript.

4.     Lines 135-138, the statistical analyses of microbiome should be written in lines 178-184 (2.6. Statistical analyses).

5.     Line 31, “sNPF” should be italicized when it represents transcripts.

6.     Line 162, “eantiserum” should be “antiserum”.

7.     Line 224, the proportion of Figure 3a and Figure 3b does not match.

8.     Line 289, “shrimp” should be “black tiger shrimp”.

9.     Line 416, “Aquaculture” should be italicized.

Author Response

To reviewer 1

  1. In the Introduction section, more references of brain-gut peptide should be cited. Apart from NPY family, as I know, many neuropeptides, such as AST-A, AST-B, AST-C, DH31, tachykinin and CCHamide, located in arthropod gut. The authors should cite more articles: Ren, Guilin R., et al. PloS one 10.7 (2015): e0133017.; Wu, Kai, et al. Frontiers in physiology 11 (2020): 191.; Veenstra, Jan A., et al. Cell and tissue research 334 (2008): 499-516.; Wegener, Christian, and Jan A. Veenstra. "Chemical identity, function and regulation of enteroendocrine peptides in insects." Current opinion in insect science 11 (2015): 8-13.

Response: Thank you for your valuable suggestion. We have added some references and related sentences in Introduction section. Please see lines 54-59.

  1. The Discussion is not sufficient. Especially the discussion of “brain-gut-microbiome” axis. References: Holzer, Peter, and Aitak Farzi. "Neuropeptides and the microbiota-gut-brain axis." Microbial endocrinology: the microbiota-gut-brain axis in health and disease(2014): 195-219.; Martin, Clair R., et al. "The brain-gut-microbiome axis." Cellular and molecular gastroenterology and hepatology 6.2 (2018): 133-148.; Kim, Boram, et al. "Response of the microbiome–gut–brain axis in Drosophila to amino acid deficit." Nature 593.7860 (2021): 570-574.; Lerner, Aaron, Sandra Neidhöfer, and Torsten Matthias. "The gut microbiome feelings of the brain: a perspective for non-microbiologists." Microorganisms 5.4 (2017): 66.

Response: Thank you for your valuable suggestion. We added a discussion about the brain-gut-microbiome” axis in Discussion section, please see lines 355-362.

  1. Authors should write consistently “mud crabs”, “the mud crabs” throughout the manuscript.

Response: Thank you. We have consistently written “mud crabs” throughout the manuscript.

  1. Lines 135-138, the statistical analyses of microbiome should be written in lines 178-184 (2.6. Statistical analyses).

Response: Thank you for your suggestion. We have re-written the sentences, please see lines 191-194.

  1. Line 31, “sNPF” should be italicized when it represents transcripts.

Response: Thank you. We have checked this error and italicized “sNPF” in the manuscript.

  1. Line 162, “eantiserum” should be “antiserum”.

Response: Word “eantiserum” have been changed to “antiserum”. Please see line 162.

  1. Line 224, the proportion of Figure 3a and Figure 3b does not match.

Response: We have improved Figure 3, Please see Figure 3.

  1. Line 289, “shrimp” should be “black tiger shrimp”.

Response: Thank you. We have added “black tiger” in the front of “shrimp”, please see line 307.

  1. Line 416, “Aquaculture” should be italicized.

Response: Thank you. We have corrected this mistake, please see line 504.

Once again, thank you very much for your comments and suggestions. We have re-written many sentences of the manuscript. The changes marked in red in revised manuscript.

Reviewer 2 Report

Comments and Suggestions for Authors

The study evaluated the gut microbiota composition and brain-gut short neuropeptide F (sNPF) expression under feed deprivation and starvation conditions in mud crab. However, there are many problems in the manuscript, and the experimental design has major defects. The measured indices were insufficient to support the conclusion.

1. The study designed 2 experiments. The first experiment was set with 3 feed deprivation levels, and the second experiment was set with 4 feed deprivation levels. The second experiment included the first experiment, and there was obviously some duplication of work. In addition, the the 72h or 96h feeding period is too short to thoroughly assess effects of the imposed changes in starvation.In fact, a period of at least 4 weeks was needed to obtain more reliable samples.

2. Before sampled , how was the mud crab fed during the feeding experiment? Feeding frequency? Feeding time?

3.Weight changes should be provided.

4. PCA analysis of gut microbiota at the phylum level is too broad, so PCA analysis at the genus level should be provided.

5. In Figure 2 and Figure 3, the words are too small to see clearly.

6. In figure 4, authors should provide and compare the immunohistochemical photomicrographs of brain-gut short neuropeptide F (sNPF) of all starvation groups.

7. Figure 5. Only half of the picture can be seen.

Author Response

To reviewer 2

The study evaluated the gut microbiota composition and brain-gut short neuropeptide F (sNPF) expression under feed deprivation and starvation conditions in mud crab. However, there are many problems in the manuscript, and the experimental design has major defects. The measured indices were insufficient to support the conclusion.

  1. The study designed 2 experiments. The first experiment was set with 3 feed deprivation levels, and the second experiment was set with 4 feed deprivation levels. The second experiment included the first experiment, and there was obviously some duplication of work. In addition, the the 72h or 96h feeding period is too short to thoroughly assess effects of the imposed changes in starvation.In fact, a period of at least 4 weeks was needed to obtain more reliable samples.

Response: Thank you for your suggestions. Indeed, there are overlapping between our two experiments. In the first experiment, we set up three groups to detect the effects of feed deprivation on the gut microbiota of mud crabs. The results showed significant changes in the gut microbiota of mud crabs under different feed deprivation levels. Based on this, we examined the impact of feed deprivation on the expression of sNPF. Referring to the results obtained from the first experiment, we established some overlapped experimental groups, including starvation group, 1/2 EF group, and EF group, to compare the changes in microbiota and sNPF gene expression under the same experimental conditions. Additionally, considering the sensitivity of gene expression, we added a 1/4 EF group in the second experiment to more closely detect the effect of feed deprivation on sNPF gene expression. The results indicated that in the 1/4 EF group, the changes in sNPF transcript were more pronounced in the later stages of the experiment.

Regarding the duration of the experiments, it is true that longer periods of starvation could obtain more reliable samples. However, considering that the mud crabs we selected are in the juvenile stage, and based on our observations in both aquaculture and laboratory experiment. We found that 72 or 96 hours of starvation already led to significant behavioral differences in the juvenile crabs, such as reduced activity range and sluggish movements under complete starvation. Moreover, both the microbiota and expression of sNPF exhibited significant differences and fluctuations. Therefore, we believe that the experimental settings of 72 or 96 hours are sufficient to meet our results and conclusions.

  1. Before sampled , how was the mud crab fed during the feeding experiment? Feeding frequency? Feeding time?

Response: We apologize for not clearly describing this part of the experimental procedure. We have added more details in the manuscript, please see lines 85-92.

3.Weight changes should be provided.

Response: Thank you for your suggestions. Undoubtedly, measuring body weight would enrich our experimental results. However, we did not assess the changes in body weight before and after feeding in our experiments. Nevertheless, we believe that this will not affect our results and conclusions.

  1. PCA analysis of gut microbiota at the phylum level is too broad, so PCA analysis at the genus level should be provided.

Response: Thank you. Base on your advice, we have added PCA analysis at the genus level, please see Figure 1.

  1. In Figure 2 and Figure 3, the words are too small to see clearly.

Response: We have re-adjusted these Figures and it should be clear enough this time.

  1. In figure 4, authors should provide and compare the immunohistochemical photomicrographs of brain-gut short neuropeptide F (sNPF) of all starvation groups.

Response: In this study, the results of immunohistochemistry were aimed at detecting the cellular localization of sNPF in the brain and gut, and it is a qualitative experiment that did not assess its changes in different feed deprivation experiments. Your suggestion will provide us with great ideas for our future research. Thank you.

  1. Figure 5. Only half of the picture can be seen.

Response: Sorry for the mistake, we have re-adjusted the Figure 5.

Once again, thank you very much for your comments and suggestions. We have re-written many sentences of the manuscript. The changes marked in red in revised manuscript.

Reviewer 3 Report

Comments and Suggestions for Authors

Review of …Effect of dietary restriction on gut microbiota….. Animals 2024 MDPI

 This is a very interesting study. The results are indicative of what the authors state. It is exciting to see such promising results and hopeful this report will encourage more trials in animal models for acute and chronic observations in the relationships of sNPF and the gut brain interactions.

It is an interesting speculation from the findings, and it is worth suggesting. Future research can focus on this possibility. It would be of interest if manipulating the levels of sNPF in fed and unfed animals by injection would result in behavioral changes.

Can the authors  address if any bacteria might be introduced by the food source provided? If the food was also examined for bacteria  that would be interesting to note. Perhaps the type of food consumed would also change the diversity and levels of the various forms of organisms in the GI track. Which particular food last consumed (i.e., high fat, high protein or carbohydrates ) and then starved  would have a great impact of the levels of sNPF.

So possible the authors could list more potential future experiments to address some of the topics related to the overall story and how to mechanistically address  the impact of sNPF.

Minor points

1.       Even though sNPF is in the title it should be defined in the abstract

2.       Line 78: Is the “enough food (EF)” group one that is fed enough that they do not eat any more ? So, is food left over in the tank? It is hard to know how “enough food” is determined.

3.       I would suggest having Figures 2 and 3 added to supplemental data with enlarged graphics so the text can be read or separate the parts and enlarge in the body of the text.

4. Line 80 "After 72 hour" but it should be "After 72 hours"?

5. Figure 5 need explanations to the letters above the histograms in the figure legend

Comments on the Quality of English Language

Ok , just a few edits.

Author Response

To reviewer 3

This is a very interesting study. The results are indicative of what the authors state. It is exciting to see such promising results and hopeful this report will encourage more trials in animal models for acute and chronic observations in the relationships of sNPF and the gut brain interactions.

It is an interesting speculation from the findings, and it is worth suggesting. Future research can focus on this possibility. It would be of interest if manipulating the levels of sNPF in fed and unfed animals by injection would result in behavioral changes.

Can the authors  address if any bacteria might be introduced by the food source provided? If the food was also examined for bacteria  that would be interesting to note. Perhaps the type of food consumed would also change the diversity and levels of the various forms of organisms in the GI track. Which particular food last consumed (i.e., high fat, high protein or carbohydrates ) and then starved  would have a great impact of the levels of sNPF.

So possible the authors could list more potential future experiments to address some of the topics related to the overall story and how to mechanistically address  the impact of sNPF.

Response: We are very pleased to receive your recognition. Your suggestions have been extremely helpful for our future research on sNPF as a brain-gut peptide. Based on your recommendations, we will attempt to draft a follow-up research plan. First, in the microbiome experiments, we will first assess the microbial community in the food, and then examine the impact of these microbial community on mud crabs in different feed deprivation levels. Second, we will design the feed deprivation experiments more meticulously, primarily to explore the effects of different food components on sNPF and the microbial community. Third, we will inject sNPF and observe the feeding behavior of the blue crab and any potential changes in the associated microbial community. Finally, based on the suggestion from other reviewer, we will investigate the cellular localization of sNPF in the brain and gut across different feed deprivation groups, as well as changes in its protein expression levels and cell counts. Of course, all these more in-depth follow-up experimental designs will be based on the results and conclusions of this study.

Minor points

  1. Even though sNPF is in the title it should be defined in the abstract

Response: Thank you. Base on your suggestion, we have added the definition of sNPF in the abstract, please see line 21.

  1. Line 78: Is the “enough food (EF)” group one that is fed enough that they do not eat any more ? So, is food left over in the tank? It is hard to know how “enough food” is determined.

Response: Thank you. In the pre-experiment, to determine the maximum daily food consumption, juvenile crabs (n=20) were fed on weighed chopped Manila clam, Ruditapes philippinarum to determine the maximum daily food consumption and the remaining food was removed, dried, and weighed after 24 hours. Finally, we calculated the food weight for the enough food group was 10 mg. In the formal experiment, the weight of the crab is consistent with the weight of the crab for determining the maximum daily food consumption in the pre-experiment. We have added some sentences about feed deprivation experimental details in Materials and Methods section, please see lines 85-92.

  1. I would suggest having Figures 2 and 3 added to supplemental data with enlarged graphics so the text can be read or separate the parts and enlarge in the body of the text.

Response: Thank you. We have re-adjusted the picture and it should be clear enough this time.

  1. Line 80 "After 72 hour" but it should be "After 72 hours"?

Response: Thank you. We have revised this mistake, please see line 101.

  1. Figure 5 need explanations to the letters above the histograms in the figure legend

Response: Thank you. We have added the explanations to the letters in Figure 5 legend please see lines 293-295.

Once again, thank you very much for your comments and suggestions. We have re-written many sentences of the manuscript. The changes marked in red in revised manuscript.

Reviewer 4 Report

Comments and Suggestions for Authors

I enjoyed reading this manuscript. The subject of study is a commercially important crustacean species and the theme is interesting to aquaculture development and crustacean physiology. It describes the alterations in the gut microbiota of the mud crab and also investigates the physiological functions of the gut-brain sNPF under feed deprivation and starvation conditions. The findings provide novel insights into the dynamic interplay between gut microbiota and sNPF in response to diet restriction. However, the manuscript needs to be reviewed and revised before acceptance. Therefore, I have some questions and suggestions to improve the quality of the paper.

Line 147: the format of reference “Johard et al., 2008” must be revised.

Line 172,174: “Takara” should be “TaKaRa”.

Line179-184: Please confirm, how homogeneity of variances was checked. 

How many biological and technical replicates should be stated at the materials and methods section.

Line 224: Figure 3a and Figure 3b are not indicated in the note.

Line 339-341: you should mention in which animal the olfactory bulb was studied. Has this been studied in other crustaceans?

Line 345-348: More detail should be provided for the previous studies of sNPF or NPY in the gut.

Comments on the Quality of English Language

I understand most of it, but it would be helpful to have a native speakers or professional editor check the English.

Author Response

To reviewer 4

I enjoyed reading this manuscript. The subject of study is a commercially important crustacean species and the theme is interesting to aquaculture development and crustacean physiology. It describes the alterations in the gut microbiota of the mud crab and also investigates the physiological functions of the gut-brain sNPF under feed deprivation and starvation conditions. The findings provide novel insights into the dynamic interplay between gut microbiota and sNPF in response to diet restriction. However, the manuscript needs to be reviewed and revised before acceptance. Therefore, I have some questions and suggestions to improve the quality of the paper.

Line 147: the format of reference “Johard et al., 2008” must be revised.

Response: Thank you. We have revised this mistake, please see line 158.

Line 172,174: “Takara” should be “TaKaRa”.

Response: Thank you. This word has been revised, please see lines 183-185.

Line179-184: Please confirm, how homogeneity of variances was checked. 

How many biological and technical replicates should be stated at the materials and methods section.

Response: Thank you for your suggestion. We have added related details in the Materials and Methods section, please see lines 188-189, 197.

Line 224: Figure 3a and Figure 3b are not indicated in the note.

Response: Thank you. We have added “(a)” and “(b)” in the note of Figure 3, please see lines 262-263.

Line 339-341: you should mention in which animal the olfactory bulb was studied. Has this been studied in other crustaceans?

Response: Thanks for your suggestion. We have added some sentences to discuss the olfactory lobes in other crustaceans, please see lines 367-372.

Line 345-348: More detail should be provided for the previous studies of sNPF or NPY in the gut.

Response: Thank you for your valuable suggestion. We have added some references and related sentences in Introduction and Discussion section. Please see lines 54-59, 358-359.

I understand most of it, but it would be helpful to have a native speakers or professional editor check the English.

Response: Thanks for your suggestion. We have re-written many sentences of the manuscript. The changes marked in red in revised manuscript.

Round 2

Reviewer 2 Report

Comments and Suggestions for Authors

All my comments have been addressed.